# Energy Policy as a Socio-Scientific Issue: Argumentation in the Context of Economic, Environmental and Citizenship Education

**Hagit Shasha-Sharf [1,\*] and Tali Tal [1,2,\*]**

1   Faculty of Education in Science and Technology, Technion—Israel Institute of Technology,
    Haifa 3200003, Israel
2   Samuel Neaman Institute, Technion City, Haifa 3200003, Israel
\*   Correspondence: hagitshasha@gmail.com (H.S.-S.); rtal@ed.technion.ac.il (T.T.)

**Abstract:** One goal of environmental civic education is preparing students, both as citizens and as professionals, to use effective arguments in public debates. Such debates include dominantly economic claims, which are multifaceted and rarely taught in schools. A learning unit that applied the pedagogical principles of socio-scientific issues was developed for 'Israel's Natural Gas Export Policy', a real sustainability dilemma. The study aimed to understand how pre- and in-service science teachers craft their arguments, by comparing their written reasoned opinions on the gas export debate, before and after the learning unit. Content analysis was conducted using Grounded Theory on the two groups' texts in a multiple case study design. Five reasoning rationales were found: 'Profits and Risks', 'Ethics or Ideology', 'Pragmatic Objectives', 'Evidence Base' and 'Stakeholder Motivations'. Each rationale yielded different reasoning strategies, including 'Costs/Benefits', 'the Trade-Off Dilemma' or 'Compromise', 'Compensatory Benefits' and 'Non-Compensatory Costs/Risks'. The findings show that both groups used more argument types in the post-task. The development of 'Profits and Risks' strategies, between the pre- and post-texts, shows how the teachers' arguments became more complex and decisive. These results exemplify how the SSI-focused learning unit enables learners to enhance their critical citizenship thinking, one of the cornerstones of democracy.

**Keywords:** socio-scientific issues; argumentation; decision making; economics; sustainability education; civic education; public policy





## 1. Introduction

This study explored learning about governmental public policy through a real-world case of a controversial natural gas export policy. The public debates in Israel regarding its national energy resource export policy reflect the sustainability dilemma between the long-term utilization of a non-renewable energy resource and current economic interests that involve both intra- and inter-generational distributional questions [1,2]. Two complex relationships formed the backdrop to this learning topic: the science–policy–citizenship relationship and the economy–environment nexus.

When policy decision making about the exploitation of a country's energy resources involves environmental risk assessments, this requires scientific knowledge and evidence but is also tightly linked to social and ethical issues [3–6]. This type of decision making compels professionals and decision makers (explicitly or implicitly) to grapple with social questions, when defining what constitutes an acceptable risk and who should bear the burden of the risk [5]. In addition, public policy decision-making processes involve an interdisciplinary mixture of arguments and other real-life problems, such as biased information or stakeholders' conflicting interests [7,8]. The study's overarching goal was to examine how we are preparing students to use effective and responsible arguments, both as future citizens and as professionals, who participate in public debates or in decision-making processes.

In institutionalized decision-making processes on public policy, economic claims are also injected into environmental, scientific, social and civic claims. The economic

considerations are usually based on research and practice. However, in environmental and civic-science education, there has been relatively little discussion or research on the way in which economics or economic thinking is taught. The few studies that have dealt with these learning issues reveal major theoretical disagreements [9–11]. Therefore, we suggest teaching about the natural gas export policy, using the frameworks of socio-scientific issues (SSI) and social acute questions (SAQ) [12]. SSI and SAQ provide frameworks for learning about controversies over ideologies or world views related to central scientific ideas or theories [12–16]. In this study, through an interpretive–qualitative analysis of argumentation, we explored how the framing of an economic policy issue (the export of a natural resource) as an SSI or an SAQ paves the way for teaching and learning about economic issues in environmental and civic education.

A learning unit (LU) was developed, according to environmental and civic education goals and SSI pedagogy principles, aiming to enable students to form opinions and make decisions about the Israel natural gas export dilemma, as an ill-structured problem for which there is no single answer, and to develop argumentation skills and decision-making competencies. The LU was implemented in an environmental-orientated education programs with two different groups of teachers. The research question aimed to understand the characteristics of these teachers' arguments before and after the enactment of the LU.

In so doing, this paper makes several contributions. It describes an innovative learning framework presenting a social dilemma concerning natural resource distribution in environmental, citizenship and civic science education. The use of SSI pedagogy in this study for the teaching and learning of an economic policy enabled the students to experience a real-life sustainability dilemma. Finally, it associates environmental education, economic thinking and SSI pedagogy with 'middle-range' theoretical conclusions based on our analysis using Grounded Theory. After a description of the learning topic, we present the theoretical background, the learning unit design, the research method, the findings and finally the conclusions.

*The Topic: Israel's Natural Gas Export Policy Debate*

In 2009–2010, major offshore natural gas reserves were discovered in the territorial waters of the State of Israel. These discoveries dramatically changed Israel's energy supply options and its geopolitical position after years of relying almost completely on imported fossil fuels and functioning as an 'energy island' [17]. The gas discoveries were licensed for production to foreign and Israeli energy companies and a new local natural gas market emerged. In 2011, Israel's natural gas export policy was examined by a governmental committee (GC) known as the 'Tzemach Committee'. It was charged with developing a national policy for these natural gas reserves, including an export policy. Throughout the committee's deliberations, local and international experts and stakeholders presented their views on the rights of different stakeholders, supply and demand forecasts, as well as environmental, security and defense risks. Interest groups submitted their position papers and were later invited to present their concerns in 20 public hearings. In 2012, the GC submitted its official recommendations to the government, which was followed by substantial public protest and an appeal to the Supreme Court. A partially redacted version of the public hearings was released to the public in June 2013. In response to the public outcry, the Israeli government reduced the maximum quantity allowed for export from 53% to 40% [2].

Israel's gas supply has major consequences for the local economy, including the opportunity to reduce its dependence on petroleum imports, develop a local petrochemical industry, increase the profitability of domestic companies and diversify employment. From an environmental perspective, a series of trade-offs was discussed in relation to the gas export policy. A natural gas supply for local electricity, manufacturing and public transportation was expected to improve local air quality. However, it also endangers the marine and coastal environments, which are under constant pressure from overuse in Israel.

These concerns spiral upward with the scale and speed of natural gas resource exploitation, thus raising further questions about long-term energy management and planning [1].

The issue as to whether to authorize the export of natural gas was significant for a broad range of stakeholders in Israel. It encountered strong opposition that involved contradictory and biased evidence, as well as the different value goals presented by various stakeholders. The export policy debate involved fundamental sociopolitical questions, such as who has the rights to gas production and who is entitled to enjoy the economic benefits of these discoveries [2]

Thus, the students in the LU needed to understand the implications for the economy and the environment of exporting natural gas. This required some scientific and technological background on production, extraction, transmission and distribution technologies; quantity assessments of natural gas reserves; quantities and prices in energy markets; alternative energy resources (renewable and non-renewable); and risk evaluations to marine and coastal ecosystems, as well air pollution [1].

## 2. Theoretical Background

### 2.1. Interrelations between Science, Policy and Citizenship in Environmental Contexts

The relationship between science and society in democratic societies with free-market economies has been described from the 'techno-science' perspective, which stresses their sustainability dilemmas and other ethical concerns. Sociocultural perspectives of science emphasize the social, ethical, economic and political constraints that make up a science-oriented society [18]. These science–society interrelationships raise multiple public issues that are science-related, such as COVID-19 health risks, energy management issues and ecological threats, so that both politicians and citizens need scientific knowledge [5]. Researchers point to the changes in the perspectives of scientific experts and citizens as stakeholders in such public decision-making processes [19], as well as the role of values, in addition to fact-based evidence, within environmental sciences [20].

The rationale for citizens' participation in public decision making has been evolving on multiple grounds, including the accelerating complexity of global environmental problems, which are fraught with uncertainties. The importance of civic participation in environmental decision making has been emphasized from a democratic perspective, since there can be opposition to technocratic decisions about environmental risks. Researchers have called for a transformative change in policy design toward more user-centered approaches and a culture that allows for a variety of perspectives [21].

In view of these dynamic and multifaced relationships between science, citizens and public policy making, and the current environmental challenges, the voices calling for substantial environmental civic science education are becoming even more crucial. This type of science education sees scientific literacy as a way to cultivate a scientifically knowledgeable citizenry who can take part in democratic decision-making processes of social significance [22,23].

### 2.2. Learning Controversial Content in the Intersection between Environmental and Economics Education

Researchers concur as to the importance of developing teachers' and students' background in economics and the value of environmental education (EE) in helping to understand systematic economic structures and incentives (such as market prices or government tax/subsidies) that can cause environmental damage [9–11]. Economic knowledge can also contribute to the development of learners' decision-making competencies. The aim of economics education is to empower students to be able to lead their lives consciously, partake in society and contribute to political affairs, be capable of proper judgment and act responsibly [24].

However, the nature of economic knowledge and understanding remains controversial. Some researchers promote the scientific neo-classical approach that focuses on how economic systems can find effective solutions, at a reasonable cost, to environmental

problems [10]. Others argue that the main role of economic knowledge in EE is to critique neo-classical doctrines, since EE aims to highlight the problems engendered by this type of economics [11]. This debate is linked to differences between 'neo-classical economics' and 'ecological economics' solutions to sustainability [25]. Although the field of economics has its roots in moral philosophy, economists have embraced the scientific approach, under the assumption that the field is value-free [26].

The idea of sustainable development (SD) also fuels the tension between different views of the economic knowledge needed for EE. Several researchers have explored whether SD better serves economic or conservation goals [14]. These controversies are also reflected in economic thinking, as manifested in different views of the value of nature and intra-generational justice [18].

### 2.3. Teaching and Learning about SSI and SAQ

The SSI approach emphasizes ethical and social dilemmas related to science and technology development, such as genetically modified food [15]. 'Socio-scientific energy-related issues' draw attention to the importance of introducing the social aspects of energy to science learners [16]. The emphasis in SAQs is on sociological issues (such as globalization or immigration) as well as SSIs and is designed to challenge students to consider the acuteness of these issues [12–14].

SSI teaching and learning (SSI-TL) has been shown to be beneficial for the acquisition of scientific knowledge and the development of higher-order thinking skills [27,28], as well as 'knowledge about science' and moral sensitivity. These learning outcomes are important factors for the quality of decisions made by citizens [29]. SSI pedagogy adopts approaches that are fundamentally informed by Constructivist educational assumptions [30]. The literature presents different models for SSI-TL [31–34]. In this study, we implement a model suggested by Sadler et al. (2017) that organizes SSI-TL into three core stages: (a) encountering the focal SSI and making the connections to science ideas and societal concerns, (b) engaging in science practices, crosscutting concepts and socio-scientific reasoning practices and (c) synthesizing key ideas and practices in a summary stage [33].

The SAQ literature, the other field of orientation for this study, emphasizes the need to cope with complexity, risks and interdisciplinary knowledge when dealing with ideological controversies. SAQ learning requires engaging with evidence from a variety of fields other than science, including values, economics, local and global perspectives, governance issues and a range of stakeholder perspectives [13]. The learning of both SSIs and SAQs involves grappling with doubts about scientific information as well as its social implications. The teaching demands socio-epistemological reflexivity in the construction of the knowledge taught, which is achieved by the examination of the different possible theoretical frameworks, and the links that can be established between empirical descriptions, social factors and ideologies [12,14,35,36].

### 2.4. Argumentation and Decision Making about Sustainability Socio-Scientific Issues

In this paper, we explore learning through argumentation, by considering that evidence and reason are the fundamental values of an argument that constitute the legitimacy of a statement [37]. To analyze argumentation, we apply principles developed in two different argumentation theories, Toulmin's Argument Pattern (TAP) [38] and Walton's (1996) theory of 'argumentation schemes' [39,40], in addition to the literature on SSI informal reasoning and decision making. The key practices associated with the negotiation and resolution of SSIs include socio-scientific reasoning (SSR), which covers the recognition of the inherent complexity of the SSI, examining issues from multiple perspectives, appreciating that SSIs are subject to ongoing inquiry and exhibiting skepticism when presented with potentially biased information [41]. The SSR model was further extended by including culture and ethical principles in the assessment of risk and uncertainties during the decision-making process [18]. The literature on SAQs contributes the additional framework of socio-scientific sustainability reasoning (S3R), which has six dimensions (Problematiza-

tion, Interactions, Knowledges, Uncertainties, Values and Governance) and different levels of complexity in each dimension [13].

Arguments about SSIs have also been referred to as reason-based decisions [6], since they involve a close relationship between reasoning and decision making (DM) in the formation of thoughtful opinions about an SSI dilemma. SSIs require sophisticated DM strategies, in which learners use beliefs and values when weighing advantages and disadvantages [6,42]. SSI-DM is described in the literature in several ways, including normative models [42,43]. The formal multi-criteria DM model is based on the numerical values associated with the decision options according to criteria, and the use of mathematical formulas to reach an optimal decision that represents the maximization of the value of the decision [42,44]. In the current study, this model was used as a scaffold in the learning process. In addition, researchers have pointed to different decision-making (DM) qualities associated with energy-related SSIs, which involve trade-offs, as well as the ability to weigh decision criteria and reflect on the structure of the DM processes [42].

From a citizenship point of view, in a democratic society, informal reasoning plays a crucial role in efforts to find solutions to public problems. However, its use also raises questions about the type of knowledge constructed through such reasoning and its meanings. Research on SSR is grappling with how to understand individuals' decision-making processes and the role of argumentation as a sociocultural activity [13]. This study thus examined the following research question: Which types of arguments are made by the learners to support their preferred policy before and after the unit?

### 3. The Learning Unit

#### 3.1. The Learning Unit Design According to the SSI Pedagogy Framework

The following three principles determined the design of the learning unit (LU).

- Student-Centered Activities. Learning was based on students' activities, including participating in role-play and practicing decision making scaffolded by worksheets.
- Organizing Framework. The LU was designed to implement a sequence of core activities according to an organized structure aimed at constructing a coherent learning process in three modules: (1) an introduction, (2) a role-play and (3) a decision-making exercise with a summary.
- Flexibility. A few versions of the introduction, role-play and assignments were planned for different ages and different group sizes. The variations all related to the same single learning unit, which had the same organizational structure, database and core activities.

#### 3.2. The Three LU Modules

- Module 1—Introduction. The participants first see an 'export dilemma' in the form of a pre-questionnaire and a presentation of factual background knowledge (on the gas discoveries and technological, economic, environmental and energy basics) and are given a short introduction to key concepts in economics (such as 'resource allocation' and 'free markets'). Then, the instructor talks about theories on the relationship between the environment and the economy. The instructor presents a balanced view of these theories and covers controversial issues discussed in the introduction module.
- Module 2—Role-Play of Public Debate. In small groups, the participants prepare the position paper of a stakeholder on the export debate and then present it to the class. This module is based on material taken from the papers submitted to the governmental committee (GC), and the participants examine arguments directly from the GC's original papers, supported by summaries and professional glossary sheets as scaffolds. The stakeholders' reports were selected based on three considerations: (a) their reports revealed different views and controversies related to the environmental aspects of the export dilemma; (b) the position presented in each report was explained in detail and was supported by data; (c) overall, the reports represented a balanced view on the decision options. The stakeholders represented in the role-play were private

companies, governmental organizations, different industry and civic organizations or coalitions. Table 1 lists these stakeholders and their positions with regard to the dilemma.

- Module 3—Decision Making (DM) and Summary (Synthesis). The DM exercise enables the participants to review and analyze the information gathered from the reports for use in the next stage that involves crafting their own positions. For this purpose, we used a normative model [39] as a scaffold. The participants were instructed to assign heftiness (weights) to alternative options according to various criteria (see their topics in Figure 1), use simple arithmetic calculations and assign a total score to each alternative. The decision with the highest score represented the maximization of learners' values.

**Table 1.** Stakeholders in the role-play.

| Stakeholder | | Recommendation on Export Policy |
|---|---|---|
| 1 | Private natural gas production firm | Export permit |
| 2 | The natural gas transportation and distribution company (state-owned) | Export permit |
| 3 | The governmental committee (ministerial representatives) | Limited export permit |
| 4 | Private sector chemical industry firm (representing a sector) | Preference for local use (regarded as a limited export permit) |
| 5 | The Israeli Forum for Coast Protection (civic organization) | No export facilities near the coast (regarded as an export permit under specific conditions) |
| 6 | The Chief Scientists of the Energy and Environment Ministries (government representatives) | No export (postponed export permit, at least until 2030) |
| 7 | The Israeli Forum of Energy and Ecological-Economics Association (civic association) | No export |

**Summary of Topics for Decision-Making Exercise**

1. Reliability and redundancy of energy supply
2. Incentives for further resource exploration and development
3. Economic and workforce expansion of local industry
4. Citizens' economic welfare
5. Local air quality
6. Protection of marine and coast ecological systems
7. Foreign trade and affairs
8. Energy resources for the next generations
9. Competition between natural gas producers
10. Increased state revenues in the short term

**Figure 1.** Topic list for decision-making exercises.

After the DM exercise, a personal or group summary was written as a wrap-up activity, followed by individual texts listing the reasoned positions about the export dilemma (similar to the instructions in the Introduction).

## 4. Method

*4.1. Research Approaches: Grounded Theory and Multiple Case Studies*

The research design was based on the Constructivist interpretive paradigm [45]. A multiple case study was conducted where Grounded Theory was used for the analysis of the participants' argumentative texts from two groups of teachers and two groups of students. The different settings and group characteristics justified the use of a multiple case design [46]. The first author designed the LU and was the leading instructor in all implementations. In this paper, we report on the two teachers' case studies.

*4.2. Settings and Participants*

4.2.1. Teachers' Case Study (CS1)

The participants in CS1 were environmental sciences teachers on 17 pre-questionnaires; 12 had a Master's and five had a Bachelor's degree. Their undergraduate and graduate majors were in various science or education fields, including environmental sciences, geography, biology, chemistry, science education, teaching and learning, educational measurement, evaluation and management. The teachers' group was culturally and geographically diverse and the female/male ratio was 13/4. They were all leading, experienced teachers with at least five years' teaching experience.

The LU was enacted during an eight-hour professional development (PD) workshop for 20 environmental science teachers. The workshop was part of a two-year 60-h PD program for leading teachers. All three modules of the LU were implemented. In Module 2, a group discussion was directed at reaching a consensual group decision on the dilemma. The decision-making exercise was implemented in dyads. At the end, the group held a closing discussion as well.

4.2.2. The Career Change Pre-Service Teachers Case (CS2)

The participants in CS2 were 12 pre-service teachers with academic backgrounds in various sciences and engineering (having at least a Bachelor's degree) and previous diverse work experience. Their ages ranged from 20 to 50 years old, which points to the group's heterogeneity.

The LU was enacted as part of a course on environmental education for a second career teacher education program. The LU implementation was planned for two classes of an hour and a half each, with additional individual homework activities from Modules 1 and 3. The homework activities enabled the class to cover all three LU modules, except for the decision-making (DM) exercise, which was done on worksheets, but with a major recapitulative discussion after the role-play, which served to discuss the decisions before the post-questionnaire task.

Table 2 presents the case studies, and Table 3 presents the modifications in each.

**Table 2.** The case studies.

| Case Study (CS) | CS1 | CS2 |
|---|---|---|
| Participants | Experienced teachers | Career change pre-service teachers |
| N | 20 | 12 |
| Context of learning | Professional development (PD) | Environmental education course |
| Total learning hours | 8 h | 3 h + homework |

**Table 3.** Summary of the learning unit activities, adjusted for each case study.

|  | Module 1 | Module 2 | Module 3 |
|---|---|---|---|
| CS1 | Individual position writing; background introduction | Role-play; group decision-making discussion | Decision-making exercise in pairs and a group summary; individual position writing; group summary discussion |
| CS2 | Individual position writing (electronic) as homework; background introduction in class | Role-play; group discussion on decision-making aspects and a summary discussion | Individual position writing (electronic) as homework |

*4.3. Pre- and Post-Task*

An open-ended pre- and post-task was used as an individual learning tool and for data collection. This identical open-ended question asked the participants to write down their reasoned positions as citizens on the Israeli natural gas export policy.

Before the LU, no information on the participants' prior knowledge on the local export debate was available; thus, a brief description of the natural gas export policy dilemma was provided. Three main considerations guided the wording of this description: the text had to be brief, it had to highlight the controversial nature of the export policy (hence, two opposing positions were presented), and it had to avoid possible bias toward one position (both sides were presented in a convincing manner). The description of the debate that preceded the pre- and post-task was as follows:

Natural gas reserves have been discovered near the Israeli shoreline. Currently, the gas is used primarily by the national electricity company and large industrial manufacturers. In the future, the uses of Israeli natural gas are expected to expand further to other industrial and transportation sectors. The gas is produced by companies specializing in the discovery and production of natural gas, whose activity involves huge investments with high risks.

Some argue that in order to provide an economic incentive for the further development of Israeli gas resources, the State should authorize the sale of natural gas abroad. Others argue that exporting Israeli natural gas should be banned, since it will come at the expense of future consumption in Israel.

After this brief description, the participants were asked to write about their own position on the government policy for or against the export of natural gas. To support their argumentation, they were asked to explain their considerations and justifications, counter opposing opinions and formulate rebuttals.

*4.4. Content Analysis of Pre- and Post-Task*

To analyze the participants' written answers to the pre- and post-question, we adopted qualitative–interpretative Grounded Theory research [45]. The interpretative content analysis was based on the premise that opinions are neither correct nor incorrect. The categories have emerged from the participants' arguments. The initial coding was inductive. We carried out line-by-line analysis on the smallest meaningful units in every statement that expressed one idea or argument through several cycles of analysis [47]. The analysis has developed in an iterative manner, while adding a few literature-based categories, as detailed next. The process ended when saturation was achieved, as the changes in the final findings became smaller and smaller.

The deductive analysis served as a lens to examine the different argumentation and decision-making (DM) content. We used the terminology of both the argumentation and DM frameworks. In each text, a recommended/preferred export policy was analyzed from two perspectives: as a claim (whereas the rest of the text was analyzed as a process of reasoning, containing one or more sub-arguments) and as a decision (in the presence of different policy options). In addition, we analyzed every statement as an argument if it was composed of at least one claim and one reason [37,48]. From the DM perspective, the

following elements were examined in the text: decision options presented in the text, stated positive and negative aspects of each decision and any other identifiable elements that emerged from the data [42,43].

In the following stage, we performed constant comparisons [49] of individual texts and between pre–post paired arguments to elicit the entire repertoire of arguments. The comparisons served to generate tentative assertions and yielded sets of categories suggesting which unifying themes and which types of variation could be found. These comparisons produced a set of categories describing the content of the participants' arguments according to their reasoning rationales, i.e., the logic of their justification, and reasoning strategies, corresponding to different ways of using each reasoning logic. The reasoning rationales represent different aspects of the learning subject, used by the participants in justifying their decisions about their preferred policy, while each reasoning rationale follows a different logic of justification. The reasoning strategies express the same corresponding reasoning logic, used in different ways or emphasizing different subjects or areas, thus creating different groups of similar reasoning rationale.

To increase credibility, we conducted several peer debriefing rounds with five other researchers, who were not part of the study, in several rounds of analysis. The peer debriefing was aimed at challenging the assumptions of the analysis until a consensus on the categorization and the coding was achieved. When differences in the interpretations emerged, a discussion was conducted leading to either an agreement or an additional round of analysis. This process included the development of working definitions for the categories, thus enabling further analysis until the analysis was complete. In the final stage, the final interrater agreement for reasoning types was high (86%). Figure 2 summarizes the content analysis process.

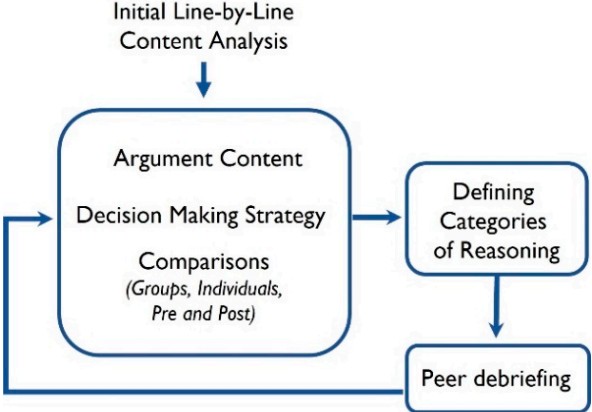

**Figure 2.** The analysis process.

Finally, after the content categories were determined, we compared the pre- and post-texts in each case study. As shown in Table 4, in CS2, the overall number of respondents to the pre- and post-task was similar, thus making it possible to draw conclusions about changes between pre- and post-arguments on a group level, even though pairing the arguments was impossible due to the anonymity requirement. In CS1, the number of responses to the pre- and post-tasks varied, but we were able to pair 18 texts (by asking the participants to add a personal code to allow for pairing of their pre- and post-answers).

**Table 4.** Pre-/post-texts in each CS.

|  | CS1 | CS1 Paired | CS2 |
|---|---|---|---|
| Pre | 17 | 9 | 10 |
| Post | 12 | 9 | 9 |
| Total | 29 | 18 | 19 |

Some of the findings were quantified to demonstrate major differences or trends. However, this does not imply that the results are generalizable.

## 5. Findings

The first section presents a characterization of all the arguments (both pre- and post-texts from both case studies) describing the types of arguments made by the learners regarding their preferred policy. The other two sections compare the pre- and post-texts in terms of the types of arguments made by the participants before and after the unit.

### 5.1. The Argument Characteristics: Reasoning Rationale and Strategies

The participants' arguments were based on their individual evaluations of policy implications, and on their choice of policy options. Their arguments, written as explanations for a preferred policy from their point of view as citizens, were categorized into reasoning rationales and strategies.

Each reasoning rationale represents a different logic of justification. Five logics emerged: 'Profits and Risks', 'Ethics or Ideology', 'Pragmatic Objectives', 'Evidence Base' and 'Stakeholder Motivations'. In addition, we found distinctive categories reflecting the differences between arguments, but using the same reasoning rationale. These differences are described as different strategies for each of the five reasoning rationales. Each argument was a response usually based on several reasoning rationales and a combination of different strategies that led to complex reasoning, which are summarized in Table 5.

**Table 5.** Reasoning rationales and strategies.

| Reasoning Rationale | Reasoning Strategies | Explanations | Quotes |
|---|---|---|---|
| Profits and Risks | Expected Benefit | Explaining only the benefits or advantages of a certain policy to justify it. | 'In my opinion, the State of Israel should approve the export of natural gas . . . because it will result in the economic development of the natural gas reserves and create employment and competition . . . ' (G1, 3Post)[1] |
|  | Expected Costs | Explaining only the costs or risks of a certain policy to justify it. | 'In my opinion, the State of Israel should not export, because it is a depletable resource and the State of Israel has few natural resources . . . also because the State is in conflict with its neighboring counties, therefore it should preserve its resources for its own needs' (G1, 2Pre)[1] |
|  | Trade-Off Dilemma | Considering the costs and benefits of policies in an unsolved manner, with no statement of the preferred policy. | ' . . . If the State of Israel starts to export natural gas . . . it will boost the Israeli economy, and, in addition, it might reduce the taxes collected by the government from the citizens for natural gas consumption. On the other hand, natural gas export will cause development . . . and a preference for export over local consumption, will cause the vast destruction of the marine ecological system and environmental pollution, as a result of natural gas production activities . . . ' (G1, 3Pre)[1] |
|  | Trade-Off Compromise | The preferred policy is presented as a compromise between contradicting consequences, and the result of obtaining some of the possible benefits while decreasing the expected losses. | 'I recommend approving natural gas export, but with a cap on quantity, because there is a need for economic growth, but on the other hand [there is a need] to protect the energy supply of the country. I will convince my colleagues, who claim to approve everything [total exports], using the argument of endangering the next generations' energy reserves. . . . [although] we have already destroyed their environment as a result of pollution, even causing the extinction of biological diversity ' (G1, 6Post)[1] |

**Table 5.** *Cont.*

| Reasoning Rationale | Reasoning Strategies | Explanations | Quotes |
|---|---|---|---|
| | Compensatory Benefits | The preferred policy is presented as beneficial, although it also includes costs and risks. The justification is based on the assumption that the expected benefits can compensate for the expected costs. | 'In my opinion, the state should approve natural gas export, but a restricted amount, that would enable out wellbeing but also the lives of the next generations . . . there are opposing claims about the natural gas being depletable, that it might run out, but I didn't accept this claim, because while we export, we can invest part of the revenues from natural gas in the discovery and development of other resources . . . ' (G1, 1Post) [1] |
| | Non-Compensatory Risks | The argument rejects a policy because of the losses and risks. This justification assumes that there are irreversible risks. | 'In my opinion, the State of Israel should ban natural gas export . . . gas export will cause price increases for the local consumers, and the competition, the race, to produce additional [natural] gas as fast as possible will lead to huge environmental hazards and dangers, which we won't be able to fix. Approving export, even if limited, would be irresponsible, based on the short-term view, since it would endanger the economy and the environment together . . . (G1, 7Post) [1] |
| Ethics or Ideology | Distributional Justice | The justification is based on the principle of distributional justice, using an ideological rule or a guiding value. | ' . . . The state economy, always, but always, takes advantage of middle-class citizen[s], and not tycoons and monopolies, which earn millions . . . ' (G1, 4Pre) [1] 'In my opinion, there is no painless way to do without a necessary and crucial resource for Israel, such as an energy resource, we can only let private firms make more profits . . . the question is: how much [should they] earn? Are the profits being reasonable or are we dealing with greed ' (G2, 3Pre) [1] |
| | Rights Protection | The justification is based on legal or normative rights protection. | ' . . . because Israelis have the right to breathe clean air and need to preserve a greener environment . . . ' (G1, 6Pre) [1] ' . . . my main counter argument is the fact that private companies discovered and produced the [natural] gas, and [so] they have the right to decide what to do with it and who to sell [it] to . . . ' (G2, 2Pre) [1] |
| | Public Decision-Making Norms | The justification is based on a normative principle for decision making. | ' . . . precautions should be taken as regards the amount of [natural] gas designated for export' (G2, 4Post) [1] 'Preserving [natural gas] reserves for more than ten years exceed a reasonable planning period[.] Planning the energy market should be made for a reasonable period of ten years ahead' (G2, 8Post) [1] |
| Pragmatic Objectives | Practical Goals | The justification is based on setting out (or defining) practical goals. | 'Approve [the export] with limitations[.] It should be produced first for the citizens . . . [we need to] think about foreign relations, provide [natural] gas to our neighbors, to get other resources from them . . . to manage the [natural] gas for longer . . . a certain amount should be preserved and not used it, for the sake of the next generations' (G1, 2Post) [1] |
| | Complementary Terms | The justification is based on specific terms, and without these terms, the justification becomes invalid. | ' . . . to export the [natural] gas and limit the exports . . . on condition that part of the profits from the exported [natural] gas is dedicated to development and incentives, so new ways to make alternative green energy available' (G2, 4Post) [1] |

| Reasoning Rationale | Reasoning Strategies | Explanations | Quotes |
|---|---|---|---|
| Evidence-Based | Evidence for Consequences and Evaluations | A notion of evidence, justifying or strengthening the justification base. | '... assuming that the [natural] gas quantity is around 1200 BCM and the State of Israel needs 600 BCM, thus half the amount needed ... (G2, 4Post) [1] |
| | Evidence-based problems: uncertainty or access to problems or biased information | A notion of problems in the evidence base, when such problems justify or strengthen a decision on a certain policy. | '... I have no idea about the state of the [natural] gas market and the gas reserves around the world ... in general, I haven't got enough information ...' (G2. 6Pre) [1] 'Currently, there are only estimates, that differ according to the institute that provides them. We have no way of knowing conclusively how much [natural] gas there is to supply and whether these amounts can meet the energetic demands of the Israeli citizens' (G2, 3Post) [1] 'The arguments of my opponents will use the extremist projection (scenario), which presents huge gas reserves as compared to the low demand for [natural] gas in the country [Israel] to convince [others] that there is no need to preserve larger reserves than the current demand' (G1, 7Post) [1] |
| Stakeholders' Motivations | Considering stakeholders' motivations | The justification is based on different stakeholders' motivations. | 'The rebuttal to my previous argument is that firms' strategic economic considerations, as stakeholders, have an interest in exporting [natural] gas, to maximize their profits [.] In addition, they [the companies] argue that in order to produce gas, they have to export it, to finance the production costs. I reject these arguments, because the state can pay these firms to preserve the gas in Israel' (G2, 5Post) [1] |
| | Representing citizens' interests | Notion of the claimant's perspective as a citizen, as an inherent part of the justification. | 'As a representative of the Israeli citizens, my position is that ...' (G2, 2Post) 'I believe that for the good of the state's citizens, it is important to ...'(G2. 1Post) |

[1] G1 indicates the teachers' group and G2 indicates the career change pre-service teachers' group. Participants are identified by number and Pre/Post indicates before or after the learning unit.

The five reasoning rationales were as follows.

- Profits and Risks—The participants explain which profits and risks are expected as outcomes of a certain policy. Their explanation of expected profits is an expression of a qualitative cost–benefit analysis, using different cost–benefit strategies. The participants also took different approaches towards compensatory risks and non-compensatory risks. The 'Profits and Risks' type of reasoning included different levels of reasoning strategies, reflecting different levels of complexity. The complex strategies, including trade-offs, were 'compromise', 'accept compensatory benefits' or, alternatively, 'reject non-compensatory risks'.

- Ethics or Ideology—In this type of reasoning, an ethical or ideological principle is used as a normative presumption. These ethical or ideological principles had to be assumed, explicitly or implicitly, to establish a valid argument. The main principles found in the arguments were distributional justice, rights protection/defense and normative principles in public decision making (such as environmental protection or responsibility for the long term).

- Pragmatic Objectives—This type of reasoning is based on the practical objectives or complementary terms needed in the framework of the policy. Not all practical considerations could be used as reasoning logic, only when the technique or type of policy implementation determines whether the policy is legitimate or not.

- Evidence Base—The evidence is couched in reasoning logic, in two different contexts. (1) References to evidence, to justify claims about a policy's implications. In these cases, the participants tapped into different types of knowledge as evidence—for example, data about the natural gas supply, or a scientific theory of depletable resources. (2) References to problems related to the evidence and affecting decision making about a policy. These included a lack of access to information, the use of biased information or references to uncertainty as an inherent element of the policy design.
- Stakeholders' Motivations—Reasons based on the motivations or duties and rights of stakeholders. The following stakeholders were mentioned in the different arguments: citizens or residents, the State, the government, private natural gas production companies, industrial or commercial firms in different sectors or markets, foreign countries and the next generations.

### 5.2. Reasoning Patterns before and after the Learning Unit

A reasoning pattern was defined as the extent to which each individual used all five reasoning rationales, i.e., 'Profits and Risks', 'Ethics or Ideology', 'Pragmatic Objectives', 'Evidence Base' and 'Stakeholders' Motivations'.

A comparison of the pre/post reasoning patterns shows that both groups used more argument types on the post-test, as depicted in Figure 3. Ethical or ideological reasoning in the CS1 teachers' group increased from 59% to 83%. The use of 'Stakeholder Motivations' rationales increased from 53% to 92%; the use of the 'Profits and Risks' rationale rose from 59% to 75%, and the use of 'Evidence-Based' rationales increased from 24% to 50%. The pre- to post-arguments also increased, as shown in in the CS1 paired arguments (9 participants), as seen in part (b) of Figure 3.

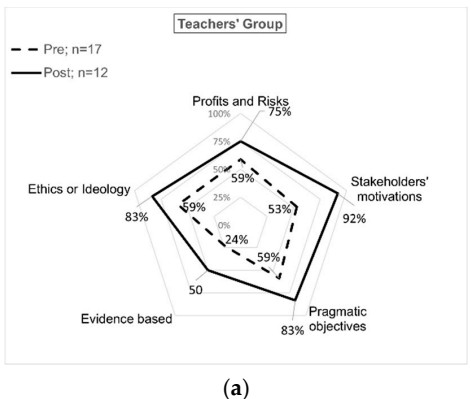

(**a**)

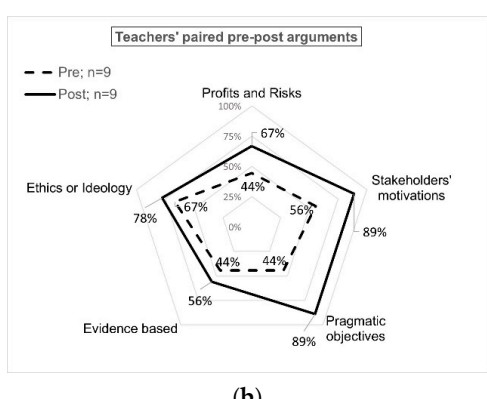

(**b**)

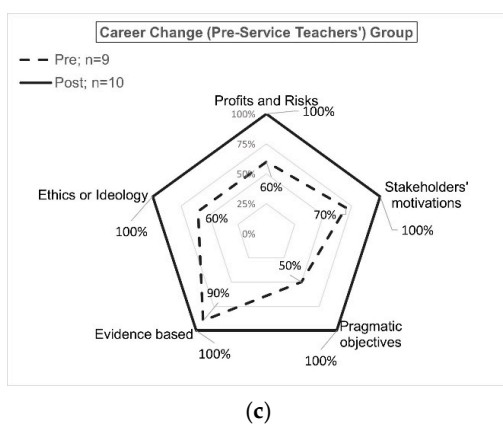

(**c**)

**Figure 3.** Comparison of reasoning scope from pre- to post-task by groups of arguments: (**a**) all arguments in the teachers' group; (**b**) paired pre- and post-arguments in 9 teachers; (**c**) all arguments in the career change pre-service teachers.

The patterns seen in the career change pre-service teachers' group (CS2) (part c) showed a substantial increase to 100% in all five reasoning rationales in the post-arguments (for example, in the pre-task, 60% addressed 'Profits and Risks' and 50% addressed 'Pragmatic Objectives').

Figure 3 also points to differences between the groups before the LU. Whereas, in the teachers' group (CS1), there was a tendency to use 'Practical Objectives', 'Stakeholders' Motivations' and 'Ethics or Ideology', the career change pre-service teachers showed a strong inclination towards 'Evidence-Based' reasoning. Despite these different perspectives, a broader scope was evident in both at the end of the LU.

### 5.3. The Pre/Post Distribution of Strategies Used in the 'Profits and Risks' Reasoning Category

The 'Profits and Risks' category that emerged in the content analysis can be summarized according to its different levels of complexity. The simplest argumentation structure consisted of a one-sided consideration of either benefits or costs or an indecisiveness strategy of 'dilemma'. The complex structures of trade-off reasoning found in the arguments was divided into three types of cost–benefit relationships: a balanced relationship (compromise), stronger benefits (compensatory benefits) or stronger costs (non-compensatory risks). The distributions of the arguments according to these categories are presented in Figure 4, which shows the changes between pre- and post-arguments and the differences between the case studies.

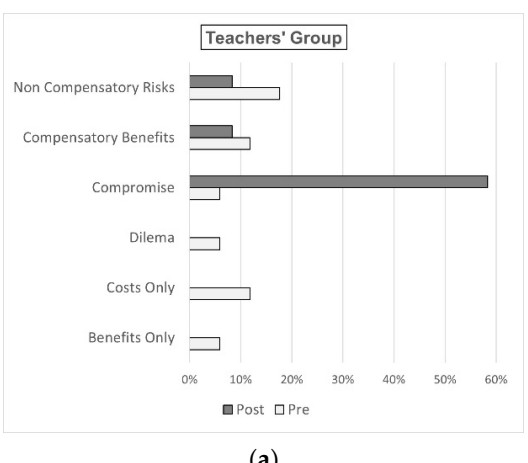

(**a**)

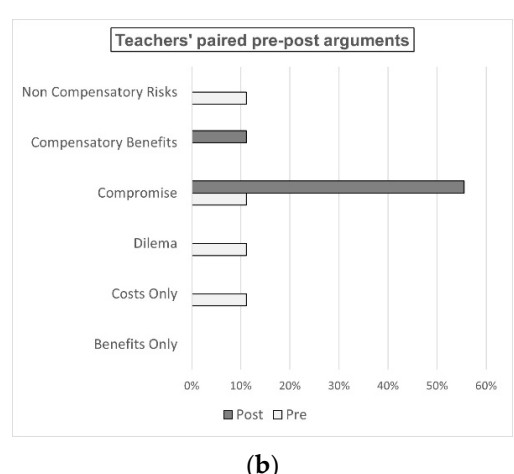

(**b**)

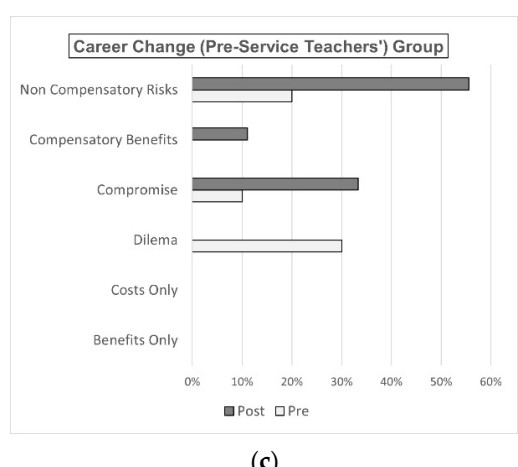

(**c**)

**Figure 4.** Comparison of pre/post distributions of different reasoning strategies classified into the 'Profits and Risks' category, by groups of arguments: (**a**) all arguments in the teachers' group (CS1); (**b**) paired pre- and post-arguments in 9 teachers in CS1; (**c**) all arguments in the pre-service teachers' group (CS2).

The 'Profits and Risks' reasoning was observed in all groups. There was an increase from pre to post, in both groups, which can be attributed to the three types of trade-off reasoning: compromise, compensatory benefits and non-compensatory costs.

Before the unit, the teachers' group used different strategies, which evolved towards the compromise strategy after the unit. The compromise strategy increased from 6% to 58% (out of a total of 78% 'Profits and Risks' arguments). In addition, 24% of the arguments written before the unit were based on simpler strategies such as one-sided considerations (either the costs or the benefits) or indecision described as a 'dilemma'. However, after the unit, the simpler strategies disappeared and were replaced by the more complex ones, including compromise, compensatory benefits and non-compensatory costs.

In the career change pre-service teachers' group, before the unit, 30% of the participants employed the dilemma strategy. After the unit, all the participants made a policy decision, with 56% arguing for loss or risk minimization and 33% choosing a compromise strategy of reasoning.

Overall, after the learning process, each group developed different reasoning strategies. While the teachers preferred to compromise, the career change pre-service teachers preferred argumentation based on non-compensatory costs. Nevertheless, both groups showed a development in their argumentation and decision making. The dilemma strategy was discarded, and both groups used more complex reasoning strategies.

## 6. Discussion

In this study, we examined the arguments written by two groups of teachers to support their positions as citizens. The arguments were produced in a learning unit on a socio-scientific issue, namely the public debate on Israel's natural gas export policy. The content analysis of the arguments examined the participants' learning outcomes, in the context of education for citizenship and sustainability.

### 6.1. Socio-Scientific and Environmental–Economic Reasoning Rationales

The participants' arguments were classified into five reasoning rationales labelled 'Profits and Risks', 'Ethics or Ideology', 'Pragmatic Objectives', 'Evidence Base' and 'Stakeholders' Motivations', with different reasoning strategies for each rationale. The five rationales are consistent with previous findings on socio-scientific reasoning, which show that they involve complexity, multiple perspectives, ethics and ideologies, as well as potentially biased information in the evidence base [41]. They also involve assessments of risks and uncertainties [16]. As noted by Simonneaux and Legardez (2010), one of the contributions of social sciences education to critical citizenship has to do with creating coherence when a whole range of possible answers is expected, by coupling them with acceptable and logical reasoning [36]. The reasoning rationales found in this study in the two different teachers' groups were written by individuals who were confronted with a real-life dilemma, characterizing their ways of resolving this controversial public issue and clarifying the decisions they made [50].

An integration of economic and environmental reasoning can be found in the 'Profits and Risks' and 'Ethics or Ideology' rationales. The 'Profits and Risks' rationale involved an analysis of policy consequences, expressed positively or negatively as benefits or costs, and uncertainties, viewed as opportunities or risks, involving economic, environmental, ethical and other consequences of the policy. Although the 'Profits and Risks' reasoning rationale reflects the links between rational decision-making processes and economic thinking, it is also indicative of SSR complexity, since it involves both environmental uncertainties and values in the participants' argumentation. Previous studies have pointed to the interplay between environmental risk assessments and value judgments [5,6,20]. Kolstø (2006) presented five types of arguments based on risk–value interactions, dubbed 'the relative risk', 'the precautionary', 'the uncertainty', 'the small risk' and the 'the pros and cons' arguments [6]. The importance of raising awareness and understanding of these interrelations between risks and values is considered to be a key issue in evaluating environmental

problems [5,6,51]. The risk–value interplay is reflected here in the 'non-compensatory risks' (representing irreversible risks) strategy, as compared to the 'compensatory benefits' strategy (representing the assumption that the risks can be compensated for). The economic–environment nexus is also reflected in the ethical reasoning rationale, corresponding to the application of an absolute principle or rule of judgment. The most common principles are related to social justice, environmental protection and protecting civil rights in governance.

The pragmatic reasoning rationale is relatively rare in relation to SSR. Our findings indicate that pragmatic reasonings described mainly how the export policy should be designed and implemented. This reasoning was only found to be valid if practical management or planning goals were presumed to be present by the arguer. Walton suggested examining pragmatic reasoning within the framework of a goal-driven claim [40,52]. A person can reason backward (rather than imagine repercussions) from the action to its necessary or sufficient conditions [40]. This reasoning rationale extended the debate to issues such as how the policy should be approved/implemented and which goals should be considered within the export policy decision making. Pragmatic reasoning creates opportunities for learners to consider goals and solutions.

### 6.2. Development of Argumentation and Learning Goals

The findings here revealed changes in reasoning strategies and patterns while learning about Israel's natural gas export policy, using an SSI approach. It was clear that engagement in the learning of a relevant dilemma and the LU tasks contributed to the development of more extensive reasoning. These findings are consistent with work by Morin et al. in that both studies found multidimensional structures of reasoning, reflected in different logics, which can be seen as different planes or dimensions [13]. In our study, the argumentation patterns reflected different levels of argument complexity, and there was a shift to higher complexity of argumentation. In order to use all five reasoning rationales in one written position, advanced integration was needed. The ability to indicate all the critical aspects, corresponding to different rules of logic, suggests multi-dimensional thinking.

Figure 5 provides examples of the leading questions associated with each reasoning rationale to further develop teaching for responsible citizenship. These questions can be used when examining a debate about public policy.

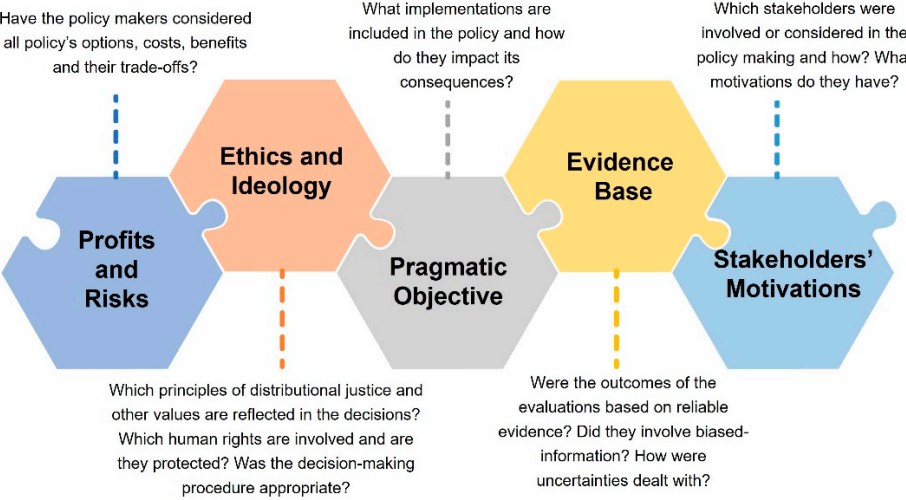

**Figure 5.** Multi-dimensional critical thinking on a public policy dilemma.

The comparison between the different 'Profits and Risks' strategies also revealed growth in argumentation skills. The 'Profits and Risks' strategies' found in the participants' arguments were benefits (only), costs (only), trade-off dilemma, trade-off compromise, compensatory benefits and non-compensatory costs/risks. These types of costs and benefits

can be organized according to their trade-off complexity. The ability to use trade-offs has been identified as crucial to developing decision-making skills on SSI [43,53]

The changes from the pre- to the post-task in the 'Profits and Risks' strategies highlight two changes: the learners' shift from simply referring to the dilemma to an argument expressing a clear decision, and the change in their reasoning strategy from one-sided consideration of either costs or benefits to strategies based on complex trade-offs. These are depicted in Figure 6. The shift to more complex strategies requires the use of knowledge about the trade-offs related to the context, and the ability to make a decision after taking the costs of such a decision into account.

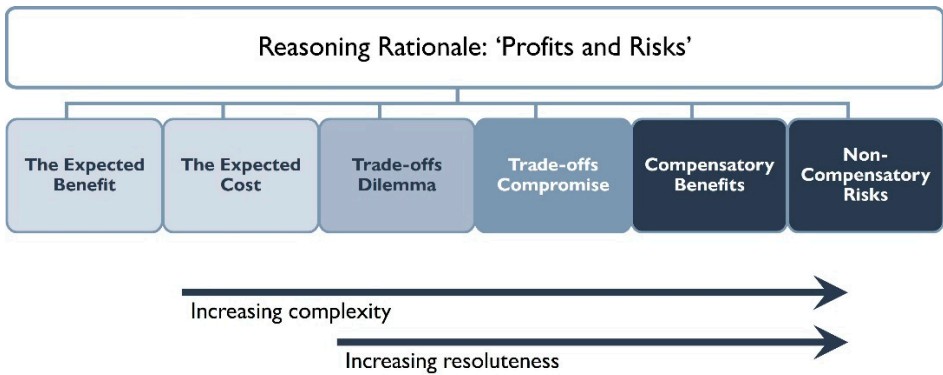

**Figure 6.** The 'Profits and Risks' reasoning strategies.

## 7. Limitations and Conclusions

The learning unit on the export of natural gas dilemma embedded key fundamental elements of environmental and economic SSI and SAQ: it consisted of a real-world controversy, with policy aspects and substantial economic and environmental implications, and required engaging in decision-making processes. There was considerable available written professional information, which is crucial for the decision-making process. Although we dealt with a local issue, it applies to many other environmental–economic issues on the use of natural resources worldwide, including taxation policies and cost–benefit dilemmas in planning, as well as investment and growth policies. In the SSI literature on sustainability issues, the economic pillar of sustainability is rarely included [12,54]. In this study, we used a real-world economic policy (a controversial export allowance) as SSI, for learning about sustainability aspects of public energy resource.

This study examined how a learning activity that (a) enabled individual participants to construct their own opinion independently, and then share with peers to reach a group decision, and (b) presented a balanced view of the different ideologies and different motivations, which can help to avoid indoctrination. The learners were able to construct their own opinions, independently, and develop their argumentation and decision-making skills [1]. We show how this development occurred, by increasing the ability to integrate different reasoning rationales, use trade-off thinking and be more decisive.

This study has several limitations. The two groups are not comparable because of the differences in participants, backgrounds and the enactment conditions, and our ability to pair pre-/post-tasks. Nevertheless, we believe that in any SSI unit, there will be different learning contexts that play a role in the teaching and learning. The findings pointed to an increase in argumentation skills in both groups. The literature suggests that different factors influence participants' arguments and decision making, such as prior knowledge, educational background and professional and personal identity [1,27–29]. Despite these factors, we found common types of changes in the participants' argumentation: the expansion of argumentation patterns and a shift to more complex reasoning strategies. Futures studies should examine the integration of economic issues into the SSI literature, with a focus on education for responsible citizenship education. These initiatives have promising potential in countering populism and developing learners' critical thinking [19].

**Author Contributions:** H.S.-S. developed the LU under the supervision of T.T. The research design was collaborative, in-cluding the task development. Instruction and data collection were done by H.S.-S. Data analysis was done collaboratively by the two authors. All authors have read and agreed to the published version of the manuscript.

**Funding:** This research received no external funding.

**Institutional Review Board Statement:** This study was approved by the Social & Behavioral Sciences Institutional Review Board (IRB) (Ethics Committee) of the Technion, approval numbers 2017-19 and 2018-3 #59931.

**Informed Consent Statement:** Informed consent was obtained from all participants involved in the study.

**Data Availability Statement:** Not applicable.

**Conflicts of Interest:** The authors declare no conflict of interest.

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
