# Peer review of "Energy Policy as a Socio-Scientific Issue: Argumentation in the Context of Economic, Environmental and Citizenship Education"

_sustainability, doi:10.3390/su15097647_

Round 1

Reviewer 1 Report

Thank you for providing me with the opportunity to review your paper. I have enjoyed reading it. However, I believe that further work is necessary before the paper is suitable for publication.

 My review will follow the format taken by the paper.

1. The introduction is too long and should present the structure of the paper.

2. An independent literature review section, following the introduction, is required. The authors should also introduce more recent references and update the literature review.

3. The conclusion has some issues. One major concern relates to the lack of academic strength of the paper. More specifically, it is not very clear what contribution your paper makes to the extant literature. And a clarification on the implications of the findings must be provided,

 4. Recheck the references and their style are according to the journal requirements, and in-text and end-text should be the same and vice versa.

I hope my feedback on this paper will help the authors to improve their work.

Author Response

Thank you for your comments.

Reviewer 2 Report

The article presented under the title “Argumentation about a controversial energy public policy, in the context of an integrated economic, environmental and citizenship education unit”, describes an interesting study in which it explored learning about a controversial public energy policy in the context of integrated economic environmental, and citizenship education.

However, there are a number of issues that recommend that this study be subjected to further review before publication. The comments included are intended for formative feedback to help authors improve the text:

- The title of the article adequately describes the experience that has been carried out, but perhaps it is too long.

- Some elements of the work summary are susceptible to improvement. First, the introduction justifying the importance of the study can be confused with the objective. Second, the methodology could have been better described; for example, study participants are not described. The description of results and conclusions, in my opinion, do not constitute 50% of the abstract (an essential requirement of this section in the journal's regulations). It would be advisable to include percentages and statistical data that support what is indicated.

- The introduction requires that research gaps be pointed out to justify the need for the study. If the gap to which it contributes is not properly uncovered, it may not make sense to do the study. It would be nice to include a heading that was on target.

- The method is the section that presents the greatest deficiencies. First, the section begins with the description of the development of the study, instead of describing the participants. Once the participants were described, it would be necessary to talk about the study design; subsequently, the procedure. After the instruments that have been used and, finally, the data analysis techniques that have been used. Well, all this is, but tremendously messy from my humble perspective. There is another issue that needs to be resolved, the results show results related to the categorization of content and, therefore, it would be convenient for the method to include the category system to carry out this categorization. Other questions arise: Was the category system created following a deductive or inductive method? How many researchers categorized the responses? Was Cohen's Kappa calculated if there were several researchers? How were the units of analysis segmented?

- Bibliography: the rules of the journal are not respected.

Author Response

Thank you for your comments.

Reviewer 3 Report

Argumentation about a controversial energy public policy, in the context of an integrated economic environmental and citizenship education unit in an interesting work however there are some issues need to be addressed before the paper ready for publication;

·       Abstract: The general background related to the topic in the beginning of abstract is missing. It should give two-to-three-line maximum background which can help let the readers easily understand what is the issue in the current research. Likewise, the purpose of the study in the abstract have simply given I suggest to arrange the abstract more logically as it simply established.

·       Introduction: The introduction begins with background however there are several statements without the support of references. I suggest to support these statements with references. Add recently published papers from 2018 and onward in the introduction. This will help shows the research gap newer. I suggest to better highlight the research innovations and gap in introduction.

·       Literature:  I still suggest to add more recent published work in a flow to better highlight the studies related to your work. I suggest few studies to be included;

https://doi.org/10.56556/jescae.v1i4.412, https://doi.org/10.56556/jescae.v1i4.397, https://doi.org/10.56556/jescae.v1i4.319, https://doi.org/10.56556/jescae.v2i1.422, https://doi.org/10.56556/jescae.v2i1.427

·       Methods: support your methodology with references and also give explanation of the suitability

·       The limitation and future research direction should be better highlighted.

Author Response

Thank you for your comments.

Reviewer 4 Report

In my opinion, the study needs to add more activity participants to assort more opinions regarding the issue. Besides, the study needs to use scientific method such the Delphi method to gather the students participants' opinions.

Author Response

Thank you for your comments.

Round 2

Reviewer 1 Report

The article was reviewed according to the proposed recommendations and meets the requirements for publication.

Author Response

Thank you for your review and your support to improve the manuscript.

Reviewer 2 Report

Good job. The article has improved a lot; however, before publishing it is necessary that they review the references section, particularly references 9, 11, 12, 22, 23, 27, 41, 46, 49, 53.

Reviewer 3 Report

7. Skilling, Barrett, Kurian, 2021. ,. Cohen, 2018. 21. 45. Taherdoost, 2023. Zeidler, Herman, Sadler, 2019.. Herman, 2018. Have been added however the author have to carefully have a look the previous review that what articles I have suggested are similar to your area and the author have to provide the reason to better highlight the gap by citing those articles in the paper or provide the reasons that how your study can be different than others in respect to the research gap.
